# Reframing Universal Lesion Segmentation as a Large-Organ Lesion Segmentation Task

Xiaoyu Bai[1,2], Ziyang Chen[1], Shaoteng Zhang[1], Qin liu[1], Xing Liu[1], and Yong Xia[1,2,3]

[1] Ningbo Institute of Northwestern Polytechnical University, Ningbo 315048, China
[2] National Engineering Laboratory for Integrated Aero-Space-Ground-Ocean Big Data Application Technology, School of Computer Science and Engineering, Northwestern Polytechnical University, Xi'an 710129, China
[3] Research & Development Institute of Northwestern Polytechnical University in Shenzhen, Shenzhen 518057, China
{bai.aa1234241, zychen, stzhang, qliu}@mail.nwpu.edu.cn, yxia@nwpu.edu.cn

**Abstract.** Universal lesion segmentation is challenging due to (1) the need to segment lesions across the entire body, often when they occupy only a small portion of the image, and (2) the crowdsourced nature of training images, leading to inconsistent annotation quality. Many existing methods adopt a divide-and-conquer strategy or integrate detection with segmentation to enhance training effectiveness. In contrast, we simplify the task by treating all lesions as a single type and directly training a universal lesion segmentation model using large image spacing and input volumes. Our approach is inspired by single-organ tumor segmentation, where including a large portion of the organ improves performance. Extending this concept, we consider the entire human body as the "organ" for universal lesion segmentation. However, applying conventional settings for single-organ segmentation to the whole body is computationally expensive and requires substantial GPU memory. To address this, we employ large volume spacing during training, effectively balancing model complexity, training cost, and performance. Our method achieved a Dice score of 0.66 and an NSD of 0.59 on the online validation set and a Dice score of 0.46 and an NSD of 0.38 on the test set, ranking first on both leaderboards. The inference time is approximately 80 seconds per case, with a GPU memory requirement of 4 GB.

**Keywords:** Universal lesion detection · Lesion segmentation · Crowdsourcing image.

## 1 Introduction

Lesion segmentation in medical images, particularly from 3D CT scans, is crucial for accurate diagnosis, treatment planning, and disease progression monitoring. Unlike single-organ lesion segmentation tasks, which focus on segmenting lesions from specific organs like the lungs or liver, universal lesion segmentation (ULS) aims to identify and segment lesions across the entire human body [27,26,30].

This problem is inherently more challenging than single-organ segmentation because:

1. It involves segmenting lesions across the entire body, where the lesion often occupies only a very small portion of the image, leading to a severe label imbalance problem.
2. The training images are crowdsourced, resulting in inconsistent annotation quality.

Widely used segmentation methods like nnU-Net [13] are typically designed for more balanced segmentation tasks, such as organ segmentation or single-organ lesion segmentation. Applying these default settings directly to ULS may yield suboptimal results. Recently, some methods have adopted a divide-and-conquer approach to convert the ULS task into multiple, easier single-organ lesion segmentation tasks [5,15], which have shown promising results. For instance, the CancerUniT method [5] decomposes the ULS task into eight single-organ segmentation and single-type lesion segmentation tasks by encoding each task as a specific query embedding. Jie Liu et al. [15] propose a CLIP-driven universal lesion segmentation model that uses text embeddings to represent different lesion segmentation tasks. While these methods effectively address challenge 1—the label imbalance problem—by restricting each sub-task's region, they introduce higher demands for data collection and labeling. Specifically, they require accurate specification of lesion types and their associated organs, which is complicated by challenge 2—the inconsistency in crowdsourced data, where label completeness and quality are often not guaranteed. For example, a dataset might only include lesion masks without specifying the lesion type, or the lesion annotations may be inaccurate.

There are also methods that treat all lesions as a single type and train a lesion/non-lesion detector [2,4]. However, these methods often train the model in a slice-by-slice 2D manner, which cannot fully leverage the 3D information of CT volumes.

Inspired by the single-organ lesion segmentation task, we propose treating the human body as a single "organ" and directly training a lesion/non-lesion segmentation model for the entire body. In single-organ lesion segmentation, achieving good performance often requires the input patch to cover a significant portion of the target organ. Similarly, to achieve optimal performance in ULS, the training image patch should cover a large portion of the entire body. Given the constraints of GPU memory, the only feasible approach is to rescale the input image to a larger spacing. Although this larger spacing is typically not recommended, as it can result in a loss of detail, we found that it works well for our ULS task. In the next section, we will give a detailed description of our method.

## 2   Method

Our method is based on the nnU-Net [13] framework, which first generates a dataset fingerprint and then automatically generates training plans accordingly.

In our approach, we made significant modifications to three key parts of the nnU-Net pipeline: preprocessing, training patch spacing and volume, and the training schedule. These adjustments led to substantial performance improvements in our experiments.

## 2.1   Preprocessing

Flare2024 task1 gives 5000 annotated CT volumes, which are gathered from multiple datasets, including:

- MSD_colon dataset
- MSD_hepaticvessel dataset
- MSD_liver dataset
- MSD_lung dataset
- MSD_pancreas dataset
- COVID-19 dataset
- KiTS23 dataset
- LIDC dataset
- TCIA-Adrenal dataset
- TCIA-LympthNodes dataset
- TCIA-NSCLC dataset
- DeepLesion dataset.

We found that some datasets include very large CT volumes, and some CT volumes contain extensive background regions, which put significant strain on I/O and CPU processing. To reduce unnecessary background, we generate a body mask by applying a threshold slightly higher than the background value and then removing the regions outside the body mask. For large CT volumes, we also divide them into two equal-sized subvolumes, as shown in Fig. 1.

Afterward, we ran the nnU-Net experiment plan function, which recommended a patch size of $96\times128\times128$, a normalized spacing of $1\times0.82\times0.82$ mm$^3$, and a batch size of 2 for a single 11GB GPU. We modified these settings by increasing the patch size to $160 \times 160 \times 160$, adjusting the normalized spacing to $1.4 \times 1.4 \times 1.4$ mm$^3$, and setting the batch size to 1 due to GPU memory limitations.

Comparing the two configurations, the original setting covers approximately a $96 \times 105 \times 105$mm$^3$ region, whereas our modified setting covers $224 \times 224 \times 224$mm$^3$. Our approach significantly increases the portion of the body covered in each patch, making the training process more efficient and effective. We then use our setting to generate the training data for our model.

## 2.2   Proposed Method

We followed the typical nnU-Net training protocol and used the default data augmentation settings. Instead of the standard U-Net model, we opted for a 3D ResU-Net, which we found to deliver better performance. Figure 3 shows the 3D ResU-Net we used.

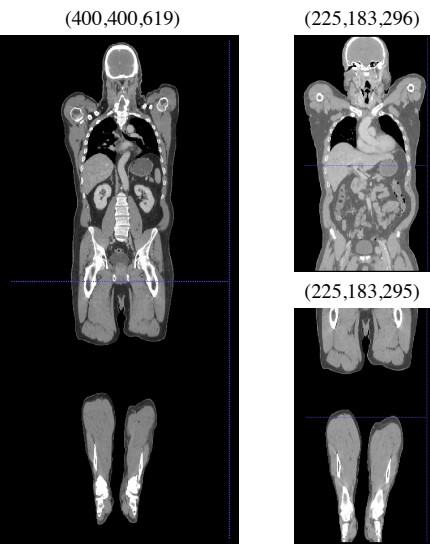

**Fig. 1.** An example of a large CT volume. The left part shows the original CT volume, with a big size of $400 \times 400 \times 619$. The right part shows the preprocessed volume, as we crop the unnecessary background and divide the original volume into two equal-sized subvolumes.

Loss function: we use the summation between Dice loss and Cross-entropy loss as the final loss function, we also use deep supervision at each downsample stage.

We did not use specific strategies to reduce the false positives on CT scans from healthy patients as false negatives are more severe than false positives for tumor diagnosis. We also did not use specific strategies to deal with partial labels. Unlabeled images and pseudo labels generated by the FLARE23 winning algorithm were also not used.

### 2.3    Post-processing

We used an overlapping sliding window approach to generate the final output for a given CT volume, with an overlap ratio of 0.5. Since the 3D ResU-Net is a fully convolutional architecture, we slightly increased the inference patch size to $192 \times 192 \times 192$, compared to the training patch size of $160 \times 160 \times 160$. We found that this approach optimizes GPU memory usage, reduces inference time, and maintains performance.

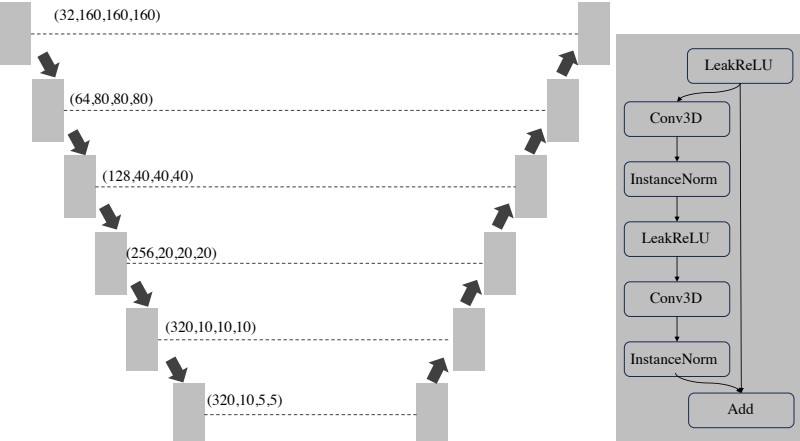

**Fig. 2.** 3D ResU-Net model. Each gray block contains two Residual blocks, as detailed in the right part of the figure.

## 3  Experiments

### 3.1  Dataset and evaluation measures

The segmentation targets cover various lesions. The training dataset is curated from more than 50 medical centers under the license permission, including TCIA [6], LiTS [3], MSD [23], KiTS [10,12,11], autoPET [9,8], TotalSegmentator [24], and AbdomenCT-1K [20], FLARE 2023 [19], DeepLesion [28], COVID-19-CT-Seg-Benchmark [18], COVID-19-20 [22], CHOS [14], LNDB [21], and LIDC [1]. The training set includes 4000 abdomen CT scans where 2200 CT scans with partial labels and 1800 CT scans without labels. The validation and testing sets include 100 and 400 CT scans, respectively, which cover various abdominal cancer types, such as liver cancer, kidney cancer, pancreas cancer, colon cancer, gastric cancer, and so on. The lesion annotation process used ITK-SNAP [29], nnU-Net [13], MedSAM [16], and Slicer Plugins [7,17].

The evaluation metrics encompass two accuracy measures—Dice Similarity Coefficient (DSC) and Normalized Surface Dice (NSD)—alongside two efficiency measures—running time and area under the GPU memory-time curve. These metrics collectively contribute to the ranking computation. Furthermore, the running time and GPU memory consumption are considered within tolerances of 45 seconds and 4 GB, respectively.

### 3.2  Implementation details

**Environment settings**  The development environments and requirements are presented in Table 1.

**Table 1.** Development environments and requirements.

| | |
|---|---|
| System | Ubuntu 18.04.5 LTS |
| CPU | Intel(R) Xeon(R) CPU E5-2690 v3 @ 2.60GHz |
| RAM | 8×16GB; 10.22MT/s |
| GPU (number and type) | Four NVIDIA RTX 2080Ti 11G |
| CUDA version | 11.8 |
| Programming language | e.g., Python 3.9.19 |
| Deep learning framework | e.g., torch 2.1.2, torchvision 0.16.2 |
| Specific dependencies | nnU-Net |

**Training protocols** We used the default nnU-Net data augmentation pipeline, which includes random rotations, random scaling, Gaussian noise, Gaussian blur, multiplicative brightness transformations, and gamma transformations applied to the input patches. Given that lesions occupy only a small portion of the entire CT volume, we employed a balanced sampling method. Specifically, each input patch has a 50% chance of containing part of a lesion. The model was trained for 2000 epochs, divided into three stages to mimic the cosine learning rate schedule. The first stage consisted of 1000 epochs. Afterward, we extended the training to 1500 epochs, continuing from the best checkpoint of the first 1000 epochs. Finally, we trained for an additional 500 epochs, initializing with the final model weights from the 1500-epoch stage. The detailed protocols are shown in Table 2.

**Table 2.** Training protocols.

| | |
|---|---|
| Pre-trained Model | None |
| Batch size | 4 |
| Patch size | 160×160×160 |
| Total epochs | 2000 |
| Optimizer | SGD |
| Initial learning rate (lr) | 1e-2 |
| Lr decay schedule | PolyLRScheduler |
| Training time | 40 hours |
| Loss function | Dice loss+CrossEntropy |
| Number of model parameters | 101.94M |
| Number of flops | 18.89T |
| $CO_2$eq | 13 Kg |

| Task: | | | Results | | FLARE24 Task 1 Evaluation | |
|---|---|---|---|---|---|---|
| # | Participant | Entries | Date | ID | Lesion DSC | Lesion NSD |
| 1 | npubxy | 1 | 2024-08-12 19:36 | 80511 | 0.66 | 0.59 |
| 2 | zhuji423 | 1 | 2024-06-06 14:02 | 67989 | 0.48 | 0.41 |
| 3 | hzhgd | 1 | 2024-07-24 15:29 | 76587 | 0.43 | 0.38 |
| 4 | taotao | 1 | 2024-07-24 15:34 | 76588 | 0.42 | 0.37 |
| 5 | stzhang | 1 | 2024-07-25 16:27 | 76800 | 0.39 | 0.27 |

**Fig. 3.** Online validation leaderboards. npubxy represents our results.

**Table 3.** Quantitative evaluation results.

| Methods | Public Validation | | Online Validation | | Testing | |
|---|---|---|---|---|---|---|
| | DSC(%) | NSD(%) | DSC(%) | NSD(%) | DSC(%) | NSD (%) |
| Algorithm1 | $36.58 \pm 30.63$ | $46.07 \pm 37.74$ | 65.51 | 59.08 | $46.46 \pm 37.13$ | $37.77 \pm 32.28$ |

## 4 Results and discussion

### 4.1 Quantitative results on validation set

On the online validation set, our method achieved a lesion Dice Similarity Coefficient (DSC) of 65.51 and a Normalized Surface Distance (NSD) of 59.08, as shown in Table 3.

**Table 4.** Quantitative evaluation of segmentation efficiency in terms of the running them and GPU memory consumption. Total GPU denotes the area under GPU Memory-Time curve. Evaluation GPU platform: NVIDIA QUADRO RTX5000 (16G).

| Case ID | Image Size | Running Time (s) | Max GPU (MB) | Total GPU (MB) |
|---|---|---|---|---|
| 0001 | (512, 512, 55) | 41.18 | 4077 | 57956 |
| 0051 | (512, 512, 100) | 25.48 | 4061 | 40665 |
| 0017 | (512, 512, 150) | 32.94 | 4077 | 68217 |
| 0019 | (512, 512, 215) | 33.90 | 4111 | 69289 |
| 0099 | (512, 512, 334) | 54.52 | 4267 | 131649 |
| 0063 | (512, 512, 448) | 33.70 | 4107 | 68803 |
| 0048 | (512, 512, 499) | 50.60 | 4087 | 85894 |
| 0029 | (512, 512, 554) | 38.68 | 4071 | 54467 |

## 4.2    Qualitative results on validation set

On the public validation set, our method achieves a DSC score of 36.58, which is significantly lower than the performance on the online validation set. We attribute this discrepancy primarily to the presence of very small lesions in the public validation set, as well as low tumor-tissue contrast. For instance, as shown in Fig. 3, the ground truth for the case LNDb-0312 consists of only one voxel, making it extremely challenging to achieve accurate predictions in such cases.

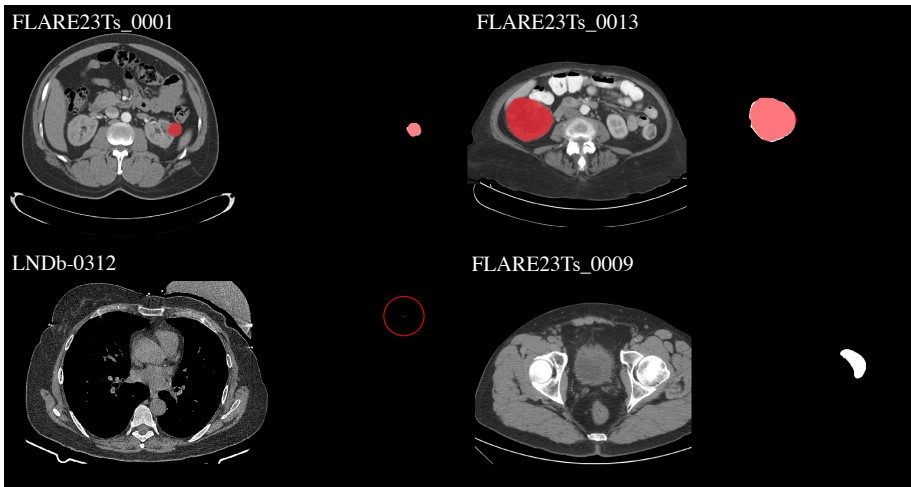

**Fig. 4.** The segmentation examples from the public validation set include two cases with good segmentation results and two with poor results. In each example, the left panel displays the CT image, while the right panel shows the ground truth in white and the predictions in red.

## 4.3    Segmentation efficiency results on validation set

Table 4.1 presents the segmentation efficiency of our method on the validation cases. Our approach requires approximately 4000 MB of GPU memory and processes each case in about 40 seconds. On the final testing set, the average processing time increases to 81 seconds per case. Additionally, on healthy CT scans, our method achieves a false positive rate of 0.15 across 40 cases.

## 4.4    Results on final testing set

On the final testing set, our method achieved a DSC of 46.46% and a NSD of 37.77% ,ranking first on the testing leaderboard. However, our approach requires approximately 90 seconds per case for inference, making it less efficient than other methods.

### 4.5   Limitation and future work

Currently, our model is trained exclusively on annotated cases, leaving a large number of unannotated cases unused due to computational constraints. Additionally, the inference time remains relatively high, primarily due to the single-process implementation in the post-processing step. In future work, we plan to leverage both annotated and unannotated cases to enhance performance and implement multi-processing in the post-processing pipeline to improve efficiency.

## 5   Conclusion

In this paper, we simplify the universal lesion segmentation task by treating all lesions throughout the body as a single category and training a universal lesion segmentation model using large image spacing and large input volumes. The rationale stems from single-organ tumor segmentation, where a substantial portion of the organ is typically used as input to achieve strong performance. We extend this concept by considering the entire human body as the "organ" for universal lesion segmentation, intentionally using large volume spacing for training. This straightforward approach effectively balances model complexity, training cost, and performance. On the online validation set, our method achieved a Dice score of 0.66 and an NSD of 0.59, securing the top position on the validation leaderboard.

**Acknowledgements**  This work was supported in part by the Ningbo Clinical Research Center for Medical Imaging under Grant 2021L003 (Open Project 2022LYKFZD06), and in part by the National Natural Science Foundation of China under Grant 62171377 and Grant 92470101. The authors of this paper declare that the segmentation method they implemented for participation in the FLARE 2024 challenge has not used any pre-trained models nor additional datasets other than those provided by the organizers. The proposed solution is fully automatic without any manual intervention. We thank all data owners for making the CT scans publicly available and CodaLab [25] for hosting the challenge platform.

## Disclosure of Interests

The authors declare no competing interests.

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

**Table 5.** Checklist Table. Please fill out this checklist table in the answer column.

| Requirements | Answer |
|---|---|
| A meaningful title | Yes |
| The number of authors ($\leq 6$) | 6 |
| Author affiliations and ORCID | Yes |
| Corresponding author email is presented | Yes |
| Validation scores are presented in the abstract | Yes |
| Introduction includes at least three parts: background, related work, and motivation | Yes |
| A pipeline/network figure is provided | Fig. 2 |
| Pre-processing | 3 |
| Strategies to use the partial label | 4 |
| Strategies to use the unlabeled images. | 4 |
| Strategies to improve model inference | 4 |
| Post-processing | 4 |
| The dataset and evaluation metric section are presented | 5 |
| Environment setting table is provided | Table 1 |
| Training protocol table is provided | Table 2 |
| Ablation study | 7 |
| Efficiency evaluation results are provided | Table 4 |
| Visualized segmentation example is provided | Fig. 3 |
| Limitation and future work are presented | Yes |
| Reference format is consistent. | Yes |