# OpenReview forum: "Reframing Universal Lesion Segmentation as a Large-Organ Lesion Segmentation Task"
_MICCAI.org/2024/Challenge/FLARE — FLARE 2024 withMinorRevisions_

### Official Review · Reviewer_Aqye · 2025-01-27
**Review of "Reframing Universal Lesion Segmentation as a Large-Organ Lesion Segmentation Task"**

**Rating:** 9
**Confidence:** 5

**Review:**

This paper presents a general lesion segmentation method based on the improved nnU-Net framework. It treats the human body as a single "organ" and addresses the problems of label imbalance and inconsistent annotation in general lesion segmentation by adjusting the preprocessing strategy (such as background removal, sub - volume segmentation), increasing the input image spacing (1.4×1.4×1.4 mm³) and training volume (160×160×160), and combining with the 3D ResU - Net model. Experiments show that this method achieves a Dice Similarity Coefficient (DSC) of 65.51 and a Normalized Surface Distance (NSD) of 59.08 on the online validation set, and performs efficiently in terms of GPU memory usage (about 4000 MB) and inference time (40 seconds per case). However, the DSC on the public validation set drops significantly to 36.58.
1. Insufficient utilization of unlabeled data
The paper clearly states that 1800 unlabeled CT scans in the training set were not used due to computational resource limitations. This may prevent the model from further optimizing its performance through semi-supervised or self-supervised learning. Especially when the labeled data is limited, the unlabeled data may contain potentially valuable information.
2. Inadequate verification of generalization ability
There is a significant difference in performance between the online validation set and the public validation set (DSC 65.51 vs. 36.58). Although the authors attribute this to small lesions and low contrast, they do not propose targeted improvement strategies (such as multi-scale feature fusion or contrast enhancement). In addition, the lack of verification on other independent datasets weakens the credibility of the method's generalization ability.
3. Failure to quantify the trade-off impact between resolution and detail loss
To cover a larger body area, the authors adjusted the input image spacing from 1×0.82×0.82 mm³ to 1.4×1.4×1.4 mm³, sacrificing detailed information (especially for small lesions). However, the paper does not quantify the specific impact of this adjustment on the segmentation performance through ablation experiments, nor does it explore how to balance the resolution and the sensitivity of lesion detection.

---

> ### Author Response · Authors · 2025-03-28
>
> Thank you for your valuable comments. The difference between the online validation results and the public validation results is primarily due to the presence of very small annotations in the public validation dataset. For instance, some cases contain lesion annotations as small as a single pixel, which may be either noise in the dataset or imperfect annotations. Missing these annotations results in a DSC of zero for those cases, significantly lowering the average DSC on the public validation set.
>
> In our experiments, we found that resampling the image from 1×0.82×0.82 mm³ to 1.4×1.4×1.4 mm³ does not harm performance. In fact, using larger spacing allows the model to process more content with the same GPU memory, thereby improving performance. In summary, as long as the resampled spacing is not excessively large, the loss of detailed resolution is acceptable.

---

### Official Review · Reviewer_B1gz · 2025-02-17
**Review of "Reframing Universal Lesion Segmentation as a Large-Organ Lesion Segmentation Task"**

**Rating:** 8
**Confidence:** 4

**Review:**

This paper propose a general lesion segmentation method based on the improved nnU-Net framework. This method treats the human body as a single "organ", and adjusts the preprocessing strategies (including background removal, sub-volume segmentation, etc.), increases the input image spacing to 1.4×1.4×1.4 mm³ and increases the training volume to 160×160×160, and combines it with the 3D ResU-Net model to solve the problems of label imbalance and labeling inconsistency in the process of general lesion segmentation.
Issues:
When evaluating the generalization ability of the method, the obvious validation deficiency was exposed. There is a huge gap between the performance of the online and public validation sets.

---

> ### Author Response · Authors · 2025-03-28
>
> Thank you for your valuable comments. The difference between the online validation results and the public validation results is primarily due to the presence of very small annotations in the public validation dataset. For instance, some cases contain lesion annotations as small as a single pixel, which may be either noise in the dataset or imperfect annotations. Missing these annotations results in a Dice Similarity Coefficient (DSC) of zero for those cases, significantly lowering the average DSC on the public validation set.

---

### Official Review · Reviewer_FGp1 · 2025-03-02
**Review of "Reframing Universal Lesion Segmentation as a Large-Organ Lesion Segmentation Task"**

**Rating:** 8
**Confidence:** 5

**Review:**

This paper simplifies the universal lesion segmentation task by treating all lesions as one type and training with large image spacing and input volumes. However, the model only uses annotated cases due to computational constraints, leaving a large number of unannotated cases unused, which may limit its performance improvement. Also, the significant performance gap between the public and online validation sets indicates potential instability, suggesting the method might struggle with small lesions and low-contrast tumor tissues. The comments are listed below:
(1) Please report the inference time and the GPU consumption in the Abstract.
(2) There are many typos or formatting issues, such as "Universion lesion detection" in key words, which should be "Universal".
(3) Identical references [13] and [14] for nnUNet.

---

> ### Author Response · Authors · 2025-03-28
>
> Thank you for your valuable comments.  Below is our point-to-point response.
> (1) Please report the inference time and the GPU consumption in the Abstract.
> We have added this information to the Abstract: The inference time is approximately 80 seconds per case, with a GPU memory requirement of 4 GB.
> (2) There are many typos or formatting issues, such as "Universion lesion detection" in key words, which should be "Universal".
> We have corrected various typos and formatting errors, including fixing "Universion lesion detection" in the keywords to "Universal lesion detection."
> (3) Identical references [13] and [14] for nnUNet.
> We have removed the redundant reference and ensured proper citation formatting.

---

### Official Review · Reviewer_8ytN · 2025-03-02
**Typos and style**

**Rating:** 9
**Confidence:** 5

**Review:**

Typos and style: There are many typos or formatting issues, such as
“...spacing of 1×0.82×0.82mm3”, space in front of the unit.
Unified significant figures in Table 4.

---

> ### Author Response · Authors · 2025-03-28
>
> Thank you for your valuable comments. We have revised the manuscript accordingly based on your suggestions.

---

### Decision · Program_Chairs · 2025-03-20

**Decision:**

Accept

**Comment:**

Please carefully address the reviewers' comments in the revision.

---

> ### Author Response · Authors · 2025-03-29
>
> We have addressed the reviewers' comments in revised manuscript.